# Healthcare professionals' views on how palliative care should be delivered in Bhutan: A qualitative study

**Tara Devi Laabar** [1,2]*, **Christobel Saunders**[1,3], **Kirsten Auret**[4], **Claire E. Johnson**[1,5,6]

**1** Medical School, The University of Western Australia, Perth, Western Australia, Australia, **2** Faculty of Nursing and Public Health, Department of Nursing, Khesar Gyalpo University of Medical Sciences of Bhutan, Thimphu, Bhutan, **3** Department of Surgery, Melbourne Medical School, University of Melbourne, Parkville, Australia, **4** Rural Clinical School of Western Australia, The University of Western Australia, Albany, Western Australia, Australia, **5** Monash Nursing and Midwifery, Monash University, Clayton, Victoria, Australia, **6** Australian Health Services Research Institute (AHSRI), University of Wollongong, Wollongong, NSW, Australia

\* taradevi.laabar@research.uwa.edu.au

**Data Availability Statement:** All data are in the supporting information files.

**Funding:** TDL is supported by Australian Government International Research Training

## Abstract

Palliative care aims to relieve serious health-related suffering among patients and families affected by life-limiting illnesses. However, palliative care remains limited or non-existent in most low- and middle- income countries. Bhutan is a tiny kingdom in the Himalayas where palliative care is an emerging concept. This study aimed to explore the views of Bhutanese healthcare professionals on how palliative care should be delivered in Bhutan. It is a component of a bigger research program aimed at developing a contextual based palliative care model for Bhutan. This is a descriptive qualitative study. Eleven focus group discussions and two in-depth interviews were conducted among healthcare professionals, recruited through purposeful sampling, from community health centres, district hospitals, regional and national referral hospitals, and the traditional hospital in Bhutan. The participants in this study emphasized the need for suitable palliative care policies; education, training and awareness on palliative care; adequate access to essential palliative care medicines; adequate manpower and infrastructure; and a multi-disciplinary palliative care team. Participants confirmed a socially, culturally and spiritually appropriate approach is crucial for palliative care services in Bhutan. Despite palliative care being a young concept, the Bhutanese healthcare professionals have embraced its importance, emphasized its urgent need and highlighted their views on how it should be delivered in the country. This study will help inform the development of a public health-focused palliative care model, socially, culturally and spiritually applicable to the Bhutanese people, as recommended by the World Health Organization.

## Introduction

The demand for palliative care has rapidly increased as the world's population ages and cancer and other chronic diseases accelerate [1]. Identified as a fundamental human right [2–5],

Program Fees Offset Scholarship and Herta Massarik PhD Scholarship for Breast Cancer Research. This study is a part of her PhD research program. The funding bodies were not involved in study design, collection, analysis, and interpretation of data or writing of the manuscript.

**Competing interests:** The authors have declared that no competing interests exist.

palliative care aims to relieve serious health-related suffering among patients diagnosed with life-limiting illnesses and their families including grief and bereavement support [6, 7].

However, palliative care remains limited or non-existent in most low-and middle-income countries (LMICs) where 76% of the need exists [1]. Since the 1980s the World Health Organization (WHO) has identified palliative care as a public health priority [8], but many LMICs have not yet integrated it into their public health agenda [9–11]. In 2014, the World Health Assembly emphasised that palliative care is an ethical responsibility of national health systems and an ethical duty for all health care providers [12]. While progress has been made in strengthening palliative care services, unacceptable gaps still exists, particularly in LMICs [1].

Bhutan is a small mountainous country located between the two giant nations, China and India. It has a population of 786,374 (2022) [13]. Bhutan is popularly known to the world for its concept of Gross National Happiness [14].

Since the early 1960s, Bhutan has made tremendous progress in modernization. Along with increased infrastructure and manpower, healthcare services have drastically improved and the life expectancy of its people have more than doubled from 35 years in the 1960s to 72 in 2021 [15]. However, today the country is challenged by the double burden of communicable and non-communicable diseases often contributing to poor quality of life and death among the Bhutanese people. Infectious diseases like HIV/AIDS, dengue fever and MDR-TB continue while cancer, organ failure, dementia and motor neuron diseases are increasing [16, 17]. Unique to Bhutan is the integration of traditional medicine as a part of the national health system [18]. A unit of traditional medicine is available in every hospital throughout the country.

Palliative care is in its infancy in Bhutan as it is in many other LMICs. In 2018, a home palliative care service was started at the Jigme Dorji Wangchuck National Referral Hospital (JDWNRH) where three beds were also allocated for palliative care in the oncology ward [19]. In 2020, a palliative care service package integrating traditional medicine into palliative care was launched by the Ministry of Health [20]. Concurrent to this study, the need for palliative care among Bhutanese patients with advanced illnesses and support needs for their family members were explored [21, 22]. The recent Bhutan cancer control strategy (2019–2025) reflected palliative care as an essential component [23].

Bhutan is predominantly a Buddhist country and despite being a small nation, there are varied, unique cultural traditions, beliefs and values particularly associated with chronic illness, death and dying within the country. Thus, an effective, integrated and culturally relevant palliative care service is crucial.

This study aimed to explore and understand the views of Bhutanese healthcare professionals on how palliative care should be delivered in Bhutan and what is needed to develop it. This study is a component of a bigger research program aimed at developing a suitable palliative care model for Bhutan.

## Methods

### Ethics statement

Ethical approval for this study was provided by the Human Research Ethics Committee at the University of Western Australia (RA/4/20/4990) and the Research Ethics Board of Health in Bhutan (REBH/Approval/2018/097). A formal written consent was obtained from the participants in the form of signature.

### Study design and setting

This is a descriptive qualitative study. Focus group discussions and in-depth interviews were conducted among healthcare professionals working in health facilities across the country. The

study sites included the national referral hospital in the capital city, Thimphu; the two regional referral hospitals located in the east and central Bhutan, four district hospitals, two community hospitals, two primary health centres (PHCs) and the main traditional hospital in Thimphu.

### Study population and sampling

Participants were recruited through purposeful sampling stratified by categories to ensure a heterogeneous distribution across health and medical disciplines.

### Data collection

Data collection was undertaken between May and August 2019 by author TDL, a Bhutanese registered nurse and lecturer with experience in palliative care. Eligible healthcare professionals were approached with permission from each hospital executive. Interested participants were requested to read the information form and their participation was confirmed with a written consent.

All focus group discussions were conducted within the facility premises in afternoons as this was a quieter time for clinicians. In two of the study sites there was only one eligible participant for whom in-depth interviews were conducted. The duration for focus group discussions and interviews ranged from 32.18–95.14 minutes with an average of 65.17 minutes.

All the focus group discussions and interviews were conducted by TDL using a semi-structured guide (**Table 1**) which was developed following review of relevant literature, discussions and consensus among authors. TDL directed questions and moderated the discussions. Most discussions were in English although there was occasional use of *Dzongkha*, the national language, and other local dialects. TDL, who has experiences working in some of these study sites, is fluent in these languages.

Achieving data saturation was not considered because immediate data transcription and analysis was not possible within the limited duration of field work where TDL was simultaneously collecting the qualitative and quantitative data among patients with advanced illness and their families for the broader research program. A purposive sampling strategy was used to recruit participants for interviews and focus group discussions—conducted in all the study sites, at all levels of healthcare across the country. Each focus group discussion and interview was audio recorded with prior consent from the participants. Field notes were maintained to record experiences and contextual realities. **Data analysis**

TDL listened to each recorded discussion several times and then transcribed verbatim. Where local dialects were used, careful translation was done prior to transcribing in English. For rigor and trustworthiness, recordings, especially involving local dialects, were cross checked with the transcripts by Bhutanese health/academic colleagues and, where necessary, corrections were made. The transcripts were entered into a qualitative data analysis software, NVivo, version 12. The steps for thematic analysis described by Braun and Clarke were used [24, 25]. Each transcript was read several times to fully understand the data. A line-by-line coding was generated to identify and interpret the patterns of themes. The identified themes were then reviewed to check if they matched the codes and data set. The themes were then named and relevant quotes were included against each theme [24]. Field notes were used to relate the contextual realities of each focus group discussion and interview. All co-authors read the transcripts, discussed and reached final consensus on the themes that were generated.

## Results

A total of 63 healthcare professionals participated in 11 focus group discussions and two in-depth interviews conducted in 12 facilities (**Table 2**). Three separate discussions were

**Table 1. A guide to focus group discussions and interviews.**

| | |
|---|---|
| **Introductory question** | Can we start by discussing about your experiences in taking care of patients with advanced illness and those at end-of-life? |
| **Transition question** | What are some of the greatest needs of these patients and their families? |
| **Focus questions** | What are some of the difficulties and challenges you encounter while taking care of these patients? |
| | What is your understanding of palliative care? |
| | Can we discuss about education and training needs for the healthcare professionals in palliative care? |
| | What do you feel about current staffing where you care for people who are dying? |
| | How can staffing impact palliative care delivery? (Number of staff, attitude of staff, training of staff, staff disciplines). |
| | And how about other resources and infrastructure if you are to provide palliative care? |
| | **Summarize the above discussion** |
| | Now can we discuss about essential palliative care drugs? |
| | How comfortably/rationally do you use morphine to treat chronic pain? |
| | What is the availability of morphine in this hospital? |
| | What is your understanding of the opioid policy in Bhutan? |
| | How does the policy need to be modified? |
| | **Summarize the discussion about the drugs** |
| | How can we deliver effective palliative care in Bhutan? |
| | Who all should be involved? |
| | Traditional medicine is important to many Bhutanese people. How can Drungtshos be involved in the care of terminally ill patients? |
| | How can the involvement of multiple disciplines/practices/traditions work together to improve the quality of life of these patients and their families? |
| | **Summarize the discussion about the multidisciplinary approach to PC** |
| | Can we now discuss about increasing awareness of palliative care in the general population? |
| **Summarizing question** | This research is aimed at facilitating integration of palliative care services in Bhutan. Considering your experiences and our discussions today, can you discuss how we can improve the care of patients with terminal illnesses, dying patients and their families? |
| | **Summarize the overall discussion** |
| **Concluding question** | Is there anything else that you feel we should have discussed but we didn't? if so, what is it? |

**Table 2. Categories of participants.**

| Participants (n = 63) | |
|---|---|
| **Categories** | ***n* (%)** |
| Nurses | 23 (36.5) |
| General medical doctors | 8 (12.7) |
| Medical specialists | 8 (12.7) |
| Pharmacists | 7 (11.1) |
| Physiotherapists | 4 (6.3) |
| Health Assistants | 4 (6.3) |
| *Drungtshos* | 9 (14.3) |

*Drungtsho*–Traditional Physician

conducted in Thimphu–one with the home palliative care group, one with other healthcare professionals at the national referral hospital and one with *Drungtshos* at the Traditional Hospital. The participants' age ranged from 24–55 years (mean 34.2, SD 7.6) and their years of service from 1–32 years (mean 8.8, SD 7.7). The number of participants in each focus group discussion ranged from two to eight. Although most of the participants were directly or indirectly involved in caring for patients with advanced illness, only the four participants from the home palliative care group were involved in a formal palliative care service. At the time of the study, this palliative care service was limited to advanced cancer patients only.

Participants confirmed it was urgent to develop palliative care in Bhutan. They could foresee that palliative care would be useful to them both professionally and personally.

> *There is an urgent need for palliative care service in our country and we really need to integrate it into our [health] system.* (General Doctor, P13)

> *Palliative care is needed in Bhutan because I am not thinking of patients (only) you know, it can be one of us one day. Everyone has to go through this which means everyone needs palliative care.* (General Doctor, P12).

Six major themes have resulted from the analysis of the transcripts: (i) government support for suitable policies; (ii) palliative care education, training and advocacy; (iii) availability and accessibility of essential palliative care medicines; (iv) workforce and infrastructure; (v) interdisciplinary palliative care team; and (vi) socially, culturally and spiritually appropriate palliative care. Here, these themes are discussed and supported by verbatim quotes from participants.

## 1. Government support for suitable policies

Participants emphasized the critical need for government and policymakers to be involved, including development of policies to enable integration of palliative care into the health system.

> *One thing is that we need to have policy in place. Please make sure that this really gets to the policymakers because we know that due to better healthcare our lifespan is increasing but NCDs [non-communicable diseases] are [simultaneously] increasing as well. So palliative care is very important.* (Physiotherapist, P49).

> *This care [palliative care] is going to be intensive, time consuming, and the financial aspect is also going to be a big burden. So [the] Ministry of Health should really be made aware and committed to initiate palliative care for the benefit of our patients.* (Specialist, P5).

While some participants suggested palliative care should be a separate program in the Ministry of Health, others recommended that it should be associated with the cancer control program.

> *First and foremost, we need to have a dedicated cancer program in the Ministry of Health and then palliative care should come as a major player in there.* (Specialist, P7).

Some participants also highlighted that palliative care should reach to '*the grassroots level*' (Nurse, P27), referring to the need to include rural communities.

Others emphasized the need for suitable palliative care guidelines.

*We do not have clear guidelines for [a] systematic way of caring for [palliative care] patients. So, if you could come up with some real evidence based guidelines.* (General Doctor, P12).

## 2. Education, training and advocacy

Most participants expressed the need for palliative care education, training and advocacy for everyone including healthcare professionals, policymakers and the general public.

*Palliative care, being relatively new, we have to sensitise our people. [Even] most of the health workers do not have an idea about what palliative care is all about. We should target health workers, general public and policymakers.* (Specialist, P7).

This theme is further divided into two sub-themes: (i) Education and training for healthcare professionals; and (ii) Public awareness and advocacy.

**i. Education and training for healthcare professionals.** Although some participants, particularly specialists, had substantial knowledge of palliative care, others were unclear what it exactly was, and some had not even heard of palliative care. While some participants knew that palliative care is for both patients and families, some understood it as an approach to care for cancer patients only and others thought it was specifically the end-of-life care. Many participants, across all disciplines, expressed they lacked palliative care knowledge and skills.

*We are not educated or qualified enough to provide such care [palliative care]. We are not able to manage pain appropriately and adequately. What I rely on is whatever I have learned during those college days in India where I think palliative care was clearly lacking.* (General Doctor, P12).

*Every day we come across people who require palliative care but we do not know how to take care because we are not taught.* (Nurse, P30).

Even those providing home palliative care service admitted they have limited knowledge and skills.

*We are not well trained. We only had workshop for ten days in Kerala [India] and with ten days of training we are posted in the home care group which is very interesting [laughs].* (Home palliative care group member, P35).

*We are mostly trained in [addressing] physical needs, so dealing with especially social needs is very difficult.* (Home palliative care group member, P16)

Besides pain management, participants faced challenges in providing psychological, emotional, and spiritual care due to lack of palliative care knowledge and skills.

*We clearly lack the understanding on the components of palliative care-the psychological and spiritual pain management. When we do not know about all these how can we give the service?* (General Doctor, P12).

Participants recommended palliative care modules to be introduced in nursing and post-graduate medical education in Bhutan.

*You should inculcate a module of palliative care for those who are undergoing nursing and post graduate medical training. . . . and it should be made compulsory to pass that module.* (Specialist, P2)

Some participants showed interest in specializing in palliative care, however, they still thought the Ministry of Health should first recognize its importance.

*This subject looks very interesting and very attractive to me [for specialization]. The first thing is, I think, the Ministry of Health and the highest decision making people should be convinced.* (General Doctor, P12).

While some participants considered education and training as crucial in improving their communication skills, others believed it would also change their attitude in caring for terminally ill patients.

*It was challenging even to help the family members understand the situation. Without training it was not easy [for me] to say that the patient may die any time, you know.* (Health Assistant, P54).

*Our attitude as health workers is very important and I think we need to be trained in palliative care which will change our mindset towards the care of very ill and dying patients.* (General Doctor, P14).

Participants also emphasised the importance of having adequate staff, especially nurses, who are trained and motivated in providing palliative care.

*For 20 to 24 hours, patients are with the nurses and 70 to 80% of patient care depends on the nurses. So, unless we have adequate nurses who are trained, motivated and dedicated . . .because at the end of the day we have to rely on nurses.* (Specialist, P5).

Some participants emphasized the need to train healthcare professionals across all levels of healthcare, including the primary health centres (PHCs) in the community where many people prefer to die.

*[Most of] our patients would ultimately want to die at home. If the patient has come [to Thimphu] from a far-flung area, like from remote eastern Bhutan, he would like to go back [home] and die. It will not be a dignified death for him when he is dying in a place which is quite foreign to him. That's why I think we should actually train healthcare workers in palliative care till the level of PHCs.* (Pharmacist, P46).

Participants who had attended palliative care workshops in the past reported they provide better care to their patients.

*After attending the workshop, I do better pain assessment, I report to the physician in a better way and patients get a better service. I get to address their problems in a holistic way.* (Nurse, P37).

*Earlier, I did not have any idea about palliative care, and we mainly focused on [nursing] procedures but now I have learned that palliative care includes all domains like physical symptom management, psychosocial and spiritual care.* (Home palliative care group member, P36).

Even participants who had never heard of palliative care in the past, strongly felt that education and training would enable them to provide better care.

*Educate us whereby our approach to care for such patients and families will improve and [then] we can improve their quality of life.* (Health Assistant, P52)

**ii. Public awareness and advocacy.** Participants discussed that it is equally important to create awareness about palliative care among the public. Some recommended schools as ideal places to create awareness on palliative care.

*The public should know about the availability of such care [palliative care] when they are struck with a terminal illness.* (Health Assistant, P54).

During the discussion, participants highlighted that most of the time their patients and families are not aware of the nature of their disease and its prognosis indicating the need for adequate information from the healthcare professionals.

*[Most of] our patients and their families firstly do not understand what their disease is, whether it is curable or not and why it is not curable.* (General Doctor, P14).

On the other hand, the home palliative care group reported that often families do not want the patient to know about his/her terminal diagnosis. Through their experiences, the group reported that home care in future may become challenging if families are not educated on the importance of truth telling.

*Fifty percent of the patients do not know about their diagnosis. [Their] families do not want them to know. I feel families are doing this because of the lack of knowledge. If the family members are not educated about truth telling I think in future home palliative care can be affected.* (Home palliative care group member, P16).

Some participants highlighted that terminally ill patients should be provided with correct information about their disease and its prognosis to motivate them to live a full life.

*They [patients with terminal illness] should be given right information and aah.. they have to know that even with terminal illness they can live their lives normally and they can contribute to the society and help their family.* (Physiotherapist, P48).

Some participants, however, emphasised that policy makers and program officers should be educated first.

*Hardly anyone in the Ministry [of Health] knows about palliative care. They should be educated first.* (Specialist, P4).

Participants also expressed the importance of educating other stakeholders in the community.

*We need to create awareness among spiritual leaders, local [traditional] healers and local leaders who will play big roles in palliative care.* (Specialist, P6).

## 3. Availability and accessibility of essential palliative care medicines

This theme is further divided into two sub-themes: (i) availability of medicines; and (ii) opioid regulation.

**i. Availability of medicines.**   While many participants reported adequate availability of palliative care medicines including opioids in their facilities, some expressed concern about a lack adequate analgesics to treat cancer pain.

*Cancer patients have multiple complaints. One thing is pain. It is very difficult to tackle pain. . . .actually [the] WHO says ladder wise pain management but here we don't have all the analgesics. There are only handful of medicines and with that we have to treat patients. So, the availability of the drugs is one major issue.* (General Doctor, P14).

Where supplied, participants faced challenges in obtaining additional stock of opioids in time and maintaining stock.

*Maintaining adequate stock of not only morphine but other opioid analgesics is certainly a challenge. If we run out of morphine tablets, there is just no way to get the additional supply immediately. The process is really hectic.* (Pharmacist, P42)

*At the most we have is morphine, of course that is quite OK for cancer patients, but then that is also in limited stock. So, we have to restrict [the use]. And to procure or mobilize from somewhere else is a big challenge because that needs money.* (General Doctor, P14).

Participants reported that the supply of morphine was, however, improving at the national referral hospital.

*The good news is that we are soon getting Morphine 20 mg and 15 mg slow release which will be very good for our palliative care patients.* (Home palliative care group member, P16).

On the contrary, opioids were not at all available in the PHCs.

*We just have Brufen [Ibuprofen] and Paracetamol and besides those two we do not have anything else [for pain management].* (Health Assistant, P54).

Some participants strongly emphasized the need to extend opioid supply to the PHCs for the benefit of patients in the community.

*By making these important drugs available only in the hospitals and not in the community we are creating more problems. The poor patients who live in the villages cannot afford to come to the hospitals just to get morphine. It should be available everywhere. I think the system should change this barrier.* (Specialist, P2)

*I now feel that the PHCs should at least have a minimal stock of morphine and other strong analgesics because sometimes if a patient has finished his morphine and comes to us and if by chance there are roadblocks due to heavy rainfall, you know. In such occasions we won't really be able to help the patient who may be in severe pain. We will be so helpless.* (Health Assistant, P54).

**ii. Opioid regulations.**   While most pharmacists felt that the current opioid regulation in Bhutan is flexible, a few considered the need to review it, especially if palliative care is to be institutionalized.

*The current opioid regulation is strict but that doesn't mean the access to the patients is denied. It is strict mainly to control the abuse but not to deny access to the patients. (Pharmacist, P40).*

*If possible, yes, [the narcotic regulation needs to be reviewed]. I think continuous availability of opioid analgesics becomes a cornerstone of pain management in palliative care. Having initiated palliative care [at JDWNRH] the requirement of opioid analgesics [has] actually doubled. If we do not control it could even triple, right? [asks his colleagues]. So, there are other confounding factors we need to also look at. (Pharmacist, P46).*

*When patients are prescribed opioids for two months it is dispensed for only one month from the pharmacy. I don't know why they dispense only for a month. (Home palliative care group member, P16).*

Participants pointed out that limited use of opioids for pain relief in LMICs could be due to inadequate knowledge about opioids among relevant people.

*Firstly, in developing countries I feel the [drug] regulators lack the idea on morphine. Many of them would not know the pharmacology of morphine. And number two is, the prescribers lack knowledge. We ourselves are afraid of prescribing opioids. (Specialist, P2).*

## 4. Workforce and infrastructure

Participants had mixed opinions on the adequacy of workforce and infrastructure if palliative care is to be initiated. Some participants expressed that they have adequate infrastructure and staff but need training. Others identified a need for additional core medical and nursing staff as well as allied health professionals to provide palliative care.

*Palliative care patients are chronically ill, bed ridden and dying who need constant support and monitoring. That demands a lot of time. At the moment, since we have shortage of staff including doctors and nurses in this hospital, it will be quite a challenge. (General Doctor, P9).*

*I don't think we have enough counsellors who can talk to the patients [and] give them emotional support. (Specialist, P3).*

Some participants identified a lack of physical facilities to provide adequate palliative care.

*In our hospital settings there is no separate doctor's room [in the ward]. We have to break bad news in the public places [general ward] where all other patients and families will hear. This is not only like a bad experience for me, but it can be disturbing even to other patients. (General Doctor, P14)*

*We definitely need more air mattresses to prevent pressure sore, a proper hospital bed where we can raise the head end and the foot end. Then we also need heaters for winters and air conditions [sic] for summers. (Nurse, P32).*

Some participants reported that limited infrastructure within hospitals led to challenges especially in fulfilling patients' wishes. They often had to deny patients' requests due to safety concerns.

*Some patients, at the end-of-life, prefer to do religious rituals* [which often involve lighting butter lamps and incense sticks] *in the hospital. They seek our permission, but we have to be cautious because in a setting like ours where cancer patients and pneumonia patients [on oxygen] are all kept together and it can be dangerous.* (General Doctor, P10).

At the national referral hospital, however, participants reported that the new dialysis centre (under construction) was expected to improve the quality of life of many patients with renal failure and their families.

*At the moment, patients requiring dialysis, cancer patients and even others are all kept in that single guest house. Once the new dialysis centre becomes functional there will be enough rooms for [renal] patients and their families. Other thing is we will have extra number of [dialysis] machines and we will be able to give adequate doses of dialysis, at least three times a week. (Specialist, P8).*

Some participants also highlighted that, if palliative and end-of-life care were to be provided in PHCs, the need for continuous oxygen supply and the provision for food may be necessary which was not available at that time.

## 5. Need for interdisciplinary palliative care team

Most participants discussed that palliative care provision needs to be an interdisciplinary team approach. Participants recognized that at least one palliative care specialist is needed to lead the team.

*In future we definitely need a palliative care specialist and [with him/her] there should be nurses who are trained in palliative care, nutritionists, physiotherapists, counsellors, psychologists, social workers, volunteers.* (Home palliative care group member, P16).

Participants across disciplines highlighted their roles in palliative care-each had something to offer as part of a multidisciplinary team.

*We have a bigger role to play in making them [patients] more comfortable.* (Nurse, P31).

*As pharmacists, we have a major role in palliative care.* (Pharmacist, P44).

*I see that our role in palliative care is crucial.* (Physiotherapist, P50).

*When we are ill it is the mind that we need to take care of. We can help to take care of the mind through counselling.* (*Drungtsho*, P59).

Participants discussed the important role that traditional medicine could play in palliative care, especially in improving the psychological and spiritual wellbeing of patients and their families.

*Our people have huge faith in traditional medicine. We know that psychological component is very important in palliative care, especially when they are at a terminal stage.* (Specialist, P7).

*I think it is a good idea to involve Drungtshos in palliative care as they can give psychological and spiritual healing and many patients get satisfied.* (General Doctor, P13)

*Drungtshos* reported that although they do not have medications to treat severe pain there are several traditional practices to heal the patients with advanced illness.

> *We have spa, massage-the traditional massage, and we have herbal bath, acupuncture and very soon we are coming up with yoga and meditation. All these are psychological and spiritual healing practices. So, it can add to the holistic approach which can benefit the patients. We just need to collaborate.* (*Drungtsho, P55*).

> *Traditional medicine is so much to deal with our mind and the spirituality of the person. In palliative care, our focus is not just the physical therapy but even the internal healing and that's why I feel those therapies like the yoga, massaging, and meditation will all be effective in palliative care.* (*Drungtsho, P61*).

While some *Drungtshos* clarified that traditional medicine need not be religion-based, others treated their chronic patients from a Buddhist perspective.

> *Sowa Rigpa [local term for traditional medicine] is the art and science of healing. So, there is nothing to do with religion. But in Bhutan we [often] contextualise traditional medicine as a Buddhist medicine. If we are to provide palliative care, we can always relate it to the science of healing.* (*Drungtsho, P59*).

> *In traditional medicine we categorise 104 diseases, including cancers and other chronic diseases, as a result of one's past Karma [principle of cause and effect]. So, we first emphasize on accumulating merit (tsho sa gop), then ask them [patients] to have faith and take refuge in the Kenchosum (the three jewels-Buddha, Dharma and Sangha) and we ask them to recite Sangay Menlha's (Buddha of Medicine's) mantra. They may not be able to prostrate but we ask them to make offerings and to rely on the Kenchosum more than ever before.* (*Drungtsho, P60*).

## 6. Socially, culturally and spiritually appropriate palliative care

Participants emphasized that palliative care in Bhutan should be realistic to the Bhutanese context. Having read the participant information form, many of them appreciated that the bigger research program (for which this paper is one of the components) is aimed at developing a socially, culturally and spiritually appropriate palliative care model for Bhutan.

> *We cannot bring in palliative care ideas of a developed country like Singapore or US and apply here because our culture is different, our religion is different and people's expectations are different with lot of cultural influences. I am very happy about the topic. Your project is on making palliative care from a Bhutanese perspective which means you are going to basically customize palliative care to our own setting which is very good. All of us believe that palliative care has to be customized.* (*Specialist, P7*).

> *If you can keep in mind our cultural background, our religious background and our society, I feel palliative care will be successful in Bhutan.* (*Pharmacist, P46*).

Participants reported huge socio-economic burden for patients and families with advanced illness and discussed that palliative care should address it.

> *The economic burden adds so much on to the patient's suffering. If that socio-economic factor can be reflected strongly in your model there may be lots of ways to tackle, like tying up with some donor agencies, and it can be really helpful.* (*Specialist, P7*).

Participants also identified the importance of facilitating religious and spiritual practices while caring for terminally ill patients.

*Spirituality has a big role when patients are dying. Palliative care must support spiritual practices.* (General Doctor, P10).

*Those dying patients seem to have peace if they have Lamas [Buddhist master] around to perform some prayers for them.* (Nurse, P23).

However, participants also reported that at times they faced challenges balancing between health and cultural/ religious beliefs. They were concerned about misleading local beliefs where patients were often victimised.

*The patient was in severe pain and I just wanted to relieve that pain. But then they [family members] believed that injecting the medicine is against the religion and will kill the patient. So, they didn't let me inject the painkiller.* (Health Assistant, P53)

*I have recently seen a patient who has stopped his diabetic medication for three months and all of a sudden his sugar level went so high. When asked he said he is taking Orien's medicine. This is a serious concern.* (Pharmacist, P44).

Nevertheless, participants reiterated that palliative care in Bhutan should be unique to its socio-cultural and spiritual context.

## Discussion

Despite various challenges associated with the rugged topography, scattered population and limited resources, Bhutan became a pioneer of primary health care in the region [17]. However, Bhutan is labelled among countries with no known palliative care activity [26, 27]. To achieve the sustainable development goals, the WHO reminds all countries to strengthen palliative care services [28]. Considering its cost-effectiveness and sustainability, especially for LMICs, the WHO has recommended four important public health strategies for PC; (1) suitable policies, (2) accessibility and availability of palliative care drugs, (3) education, awareness and training; and (4) implementation of palliative care services at all levels of health care [8, 29]. The WHO further recommends palliative care services to be unique to the socio-cultural and spiritual context of each country [11].

In this study, Bhutanese healthcare professionals have discussed in detail how palliative care should be delivered in Bhutan. Concurrent to the WHO recommended public health strategies, the study found that policy, education, and availability and accessibility to essential medicines are crucial to introducing palliative care in Bhutan.

National policy is a fundamental component in integrating palliative care into any health system [1, 12, 30, 31]. In Bhutan, the recently developed cancer control strategy (2019–2025) [23] has led to establishment of a cancer control program and, anecdotally, this aspires to roll out palliative care services and specifies the need to integrate it at all levels of healthcare [23]. Appropriate policies have enabled funding opportunities, facilitated better access to medicines, enhanced educational opportunities and integrated palliative care services at all levels of healthcare in Nepal, India, Tajikistan and some African countries [32–36]. With suitable policies, Bhutan can implement palliative care at all levels of healthcare. Further, in Uganda, policies have supported home-based palliative care increasing opportunities for home deaths, often preferred by patients [37]. Palliative care also supports family caregivers, reducing unnecessary hospital admissions [12, 29, 38], and increasing acute care capacity [39]. In

Bhutan, the referral hospitals are often challenged with bed shortages for acute cases because many patients with chronic conditions remain in hospital for a long time. Unlike some countries where palliative care is limited due to lack of government support [1, 40], the Bhutanese government is acknowledging its importance [20, 23]. The Health Minister confirmed the government is working towards integrating palliative care into the national health system [41].

This study emphasized the need for education, training and awareness for healthcare professionals and the public. Specific emphasis was placed on training healthcare professionals at all levels of healthcare including those working in PHCs, as well as educating policy makers. The cancer control strategy highlighted the need to train Bhutanese healthcare professionals in palliative care [23]. To date, only a handful of doctors and nurses have attended short courses in palliative care [42]. Contextually appropriate palliative care education through nursing and medical education, and as 'in-service' training programs for healthcare professionals are essential [8, 43–45]. Evidence indicates that education and training has improved knowledge and skills [46], changed attitude towards the care of very ill and dying patients [47] and enhanced access to opioid analgesics [32]. Adequately trained community health staff can provide quality palliative care at home—reducing the need for patients to travel to tertiary hospitals [30]. Recently, the Faculty of Nursing and Public Health at the University of Medical Sciences of Bhutan has developed a palliative care module for nursing students.

Although morphine and other opioids were available in all Bhutanese hospitals [48], there were often issues related to maintaining stock. Opioids were, however, not supplied to PHCs, possibly due to unavailability of doctors. In Uganda, special training for nurses and community health workers have resulted in wider accessibility of morphine for rural patients [49]. In Bhutan, where morphine was available, it was only sustained release and not immediate release formulations. Knaul and colleagues, in their Lancet Report [38], made 'emphatic recommendations' to make immediate release morphine available in oral and injectable formulations at all levels of healthcare. The interrupted supply of analgesics, particularly opioids, in Bhutan is due to inadequate policy support [23]. Educating policy makers, government leaders, drug regulators and healthcare professionals, and developing palliative care policies have drastically improved access to opioid analgesics in other LMICs [32, 33, 35, 49].

The study also identified the need for an interdisciplinary team approach for palliative care in Bhutan. Given the complexity of 'total pain' [50], palliative care is defined by a multidisciplinary approach [51]. Internationally, it is recommended that palliative care teams include doctors, nurses, physiotherapists, pharmacists, counsellors/psychologists, social workers, dieticians, spiritual leaders and volunteers [52]. While trained volunteers play an important role in palliative care offering practical and emotional support, other members' expertise are utilised as per the need of the patients and families [52]. In Bhutan, besides doctors, nurses and other allied healthcare professionals, a handful of clinical counsellors are now available in some hospitals. However, medical social workers are not available yet. With palliative care now initiated, the government should consider appointing medical social workers who could play a critical role in addressing the numerous social issues, as identified in recent evidence [52, 53]. Spiritual leaders, particularly monks at the Central Monastic Body, are becoming involved in palliative care [54].

Besides the role of medical care, the study identified a crucial role for traditional medicine. In Bhutan there are two forms of traditional medicine–official traditional medicine integrated into the national health system and the local traditional healing practices [55]. The palliative care service package launched in 2020 aimed at integrating traditional therapies, such as relaxation massage, hot and cold compression, acupuncture and mental coaching through Buddhist meditation, into palliative care [20]. Many Bhutanese also believe that diseases are caused by bad, vengeful spirits causing imbalances in bile, phlegm and wind channels within the body [56, 57]. They have strong faith in traditional healers, monks and religious practitioners who

perform special practices, rituals and ceremonies to appease bad spirits and remove impurities. Traditional healers are consulted regularly throughout the illness trajectory and they play a vital role in psychological, emotional and spiritual healing, especially at end-of-life. Traditional medicine and healing practices for chronic illnesses are also popular in other LMICs [58, 59].

Socially, culturally and spiritually appropriate palliative care was another priority described by this study. Recent studies found that financial challenges were one of the main issues faced by patients with advanced illnesses and their families in Bhutan [21, 22]. In this study, participants highlighted that the socio-economic burden encountered by patients and families must be addressed as a part of palliative care. International evidence indicates that palliative care in LMICs cannot exclude basic necessities such as food, clothing, shelter and financial assistance in addition to medical and nursing care [31, 60]. Bhutanese have unique cultural beliefs and spiritual values attached to chronic illness, death and dying. Some participants were, however, concerned about the counter-productive effects of those beliefs and values. Hence, it is crucial to involve, educate and create awareness on palliative care among the spiritual leaders, traditional healers and the local leaders. The WHO and other international palliative care organizations emphasize that PC must be socially, culturally and spiritually appropriate. Pelzang and colleagues, in their study [61], mentioned that 'cultural context' must be given due consideration if effective, sustainable and safe quality healthcare services are to be delivered in Bhutan.

## Strengths and limitations

The strength of the study is having included participants from all levels of healthcare across the country which provided opportunities to discuss palliative care at different facilities. Some of the participants had not even heard of palliative care although they had been caring for patients with advanced illness and at end-of-life. The study also helped in understanding the infrastructure and human resource realities in these facilities. However, there are also limitations to this study. Participant selection bias was inevitable because in some facilities there was just one eligible participant. For example, district hospitals had just one physiotherapist, pharmacist and *Drungtsho* and they had to be recruited by default. Similarly, at the time of study, there was only one eligible participant in one of the community hospitals and PHCs. Although some interested healthcare professionals could not participate due to their busy schedule, given the distribution of the study sites throughout the country and the diverse disciplines, the participants in this study do represent the healthcare fraternity of Bhutan.

## Conclusion

The need for palliative care will continue to increase globally, regionally and nationally. This study is the first of its kind in the tiny Kingdom of Bhutan. Despite palliative care being a young concept, the Bhutanese healthcare professionals have embraced its importance, emphasized its urgent need and highlighted their views on how it should be delivered in the country. Besides identifying the crucial need for palliative care policy, education and training, and availability and accessibility of essential palliative care drugs, particularly opioids, this study also emphasized that palliative care in Bhutan must be customized to its unique socio-cultural and spiritual context. This study will, therefore, help inform the development of a WHO-recommended public health-focused palliative care model suitable to the Bhutanese context–an objective of a bigger research program.

## Supporting information

**S1 Checklist. ISSM COREQ checklist.**
(PDF)

**S1 Data. Fieldnote.**
(DOCX)

**S2 Data. Transcript.**
(DOCX)

**S3 Data. Fieldnote.**
(DOCX)

**S4 Data. Transcript.**
(DOCX)

**S5 Data. Fieldnote.**
(DOCX)

**S6 Data Transcript.**
(DOCX)

**S7 Data. Transcript.**
(DOCX)

**S8 Data. Fieldnote.**
(DOCX)

**S9 Data. Fieldnote.**
(DOCX)

**S10 Data. Transcript.**
(DOCX)

**S11 Data. Transcript.**
(DOCX)

**S12 Data. Fieldnote.**
(DOCX)

**S13 Data. Fieldnote.**
(DOCX)

**S14 Data. Transcript.**
(DOCX)

**S15 Data. Transcript.**
(DOCX)

**S16 Data. Fieldnote.**
(DOCX)

**S17 Data. Transcript.**
(DOCX)

**S18 Data. Fieldnote.**
(DOCX)

**S19 Data. Fieldnote.**
(DOCX)

**S20 Data. Fieldnote.**
(DOCX)

**S21 Data. Fieldnote.**
(DOCX)

**S22 Data. Transcript.**
(DOCX)

**S23 Data. Fieldnote.**
(DOCX)

**S24 Data. Transcript.**
(DOCX)

**S25 Data. Transcript.**
(DOCX)

**S26 Data. Fieldnote.**
(DOCX)

## Author Contributions

**Conceptualization:** Tara Devi Laabar, Christobel Saunders, Kirsten Auret, Claire E. Johnson.

**Data curation:** Tara Devi Laabar.

**Formal analysis:** Tara Devi Laabar.

**Methodology:** Tara Devi Laabar.

**Supervision:** Christobel Saunders, Kirsten Auret, Claire E. Johnson.

**Writing – original draft:** Tara Devi Laabar.

**Writing – review & editing:** Christobel Saunders, Kirsten Auret, Claire E. Johnson.

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
