## [Decision Letter · Decision Letter 0]

6 Aug 2022

PGPH-D-22-00738

Healthcare professionals’ views on how palliative care should be delivered in Bhutan: a qualitative study

Dear Dr. Laabar,

Thank you for submitting your manuscript to PLOS Global Public Health. After careful consideration, we feel that it has merit but does not fully meet PLOS Global Public Health’s publication criteria as it currently stands. Therefore, we invite you to submit a revised version of the manuscript that addresses the points raised during the review process.

EDITOR: Thank you for submitting your manuscript to PLOS Global Health. It addresses an important topic and the reported well. I am very grateful to the reviewers for their comments. I am pleased to invite you to resubmit a revised manuscript with the following recommendations:

Please address all comments from reviewers.Please make use of the COREQ guideline (or equivalent) and attach a completed checklist: https://www.equator-network.org/reporting-guidelines/coreq/The manuscript is quite long. Can some sections in the introduction and discussion be condensed? For example, information about Gross National Happiness or indeed disease burden is not directly relevant to the topic of the paperPlease consider adding a box to summarise some key recommendations based on your findings for how palliative care can be developed in BhutanPlease do not use the abbreviation 'PC' as this is non-standard (spell out palliative care in full)

We look forward to receiving your revised manuscript.

Kind regards,

Dmitri Nepogodiev

Guest Editor

Journal Requirements:

1. In the Funding Information you indicated that no funding was received. Please revise the Funding Information field to reflect funding received.

Please ensure that the funders and grant numbers match between the Financial Disclosure field and the Funding Information tab in your submission form. Note that the funders must be provided in the same order in both places as well.

2. Please update your online Competing Interests statement. If you have no competing interests to declare, please state: “The authors have declared that no competing interests exist.”

4. We have noticed that you have uploaded Supporting Information files, but you have not included a list of legends. Please add a full list of legends for your Supporting Information files after the references list.

Reviewers' comments:

Reviewer's Responses to Questions

**Comments to the Author**

1. Does this manuscript meet PLOS Global Public Health’s publication criteria? Is the manuscript technically sound, and do the data support the conclusions? The manuscript must describe methodologically and ethically rigorous research with conclusions that are appropriately drawn based on the data presented.

Reviewer #1: Yes

Reviewer #2: Yes

2. Has the statistical analysis been performed appropriately and rigorously?

Reviewer #1: Yes

Reviewer #2: N/A

3. Have the authors made all data underlying the findings in their manuscript fully available (please refer to the Data Availability Statement at the start of the manuscript PDF file)?

Reviewer #1: Yes

Reviewer #2: Yes

4. Is the manuscript presented in an intelligible fashion and written in standard English?

Reviewer #1: Yes

Reviewer #2: Yes

5. Review Comments to the Author

Reviewer #1: Thank you for the opportunity to review this document. It is an original and significant study in the field of palliative care, in my opinion.

I have some minor suggestions:

• There is no such thing as a qualitative cross-sectional study. One of the quantitative designs is cross-sectional design. You could say: descriptive qualitative study only.

• Table 1 might be moved to the results section

• I am uncertain as to why TDL collected quantitative and qualitative data (line 28, page 6) This is a qualitative study, so please keep that in mind. Even if the researcher is collecting quantitative data for another study or for the demographic information, I do not believe this is relevant to this study's reader.

• Data analysis: please provide step-by-step instructions.

Reviewer #2: Thank you for the opportunity to review this important work. This is a lovely study and conducted and reported really well – congratulations. It will be really interesting to see your full program of research evolve – such an important program of work. I hope my comments below are helpful as you finalise this study for publication.

Abstract:

This is well written and provides a good overview of the study – well done. Just a few small thoughts for your review:

• I am not sure it is helpful to say palliative care is more necessary in one country over another to be honest – would suggest removing ‘where it is actually needed most’ and replace with ‘ despite evident need’ or something similar. Even in high-income countries there is an equal need for optimal palliative care etc.

• Line 33 – rather than palliative care ‘beginning to get recognised’ – perhaps is ‘in its infancy’ or something like that?

• Line 34 ‘their’ country

• Line 44 ‘services’

Introduction – from my perspective this provides a great introduction to your study – well written and interesting to read – well done.

Data collection – I think it would be helpful to include a table or textbox with your semi-structured question route so readers can see quickly the key elements of your discussion points (as many won’t access your transcripts in supplementary files etc).

I understand your point in relation to data saturation – but I think it would be helpful to note something about sampling that confers confidence that you considered the numbers and representatives carefully to ensure adequate information overall? You may find some useful resources to support your thinking in the following?

• Tongco MDC. Purposive sampling as a tool for informant selection. Ethnobotany Research and Applications 2007; 5: 147-158.

• Curtis S, Gesler W, Smith G, et al. Approaches to sampling and case selection in qualitative research: examples in the geography of health. Social Science & Medicine 2000; 50: 1001-1014.

• Guest G, Bunce A and Johnson L. How Many Interviews Are Enough?:An Experiment with Data Saturation and Variability. Field Methods 2006; 18: 59-82. DOI: 10.1177/1525822x05279903.

Results

Well written and easy to read. These are quite long but if they fit within the journal’s guidelines – all good. I found it a very interesting read! The quotes clearly articulate your points and support the themes noted.

Discussion

Again – well written – thank you. I just wonder if it may be helpful to frame your results within the WHO Innovative Care for Chronic Conditions Framework – it seems to really fall within their key areas strongly.

Line 471 – when you state ‘in several LMICs’ – could you be a little more specific – which LMICs and how might this be useful when considering the Bhutan context – what has worked well / what not so well etc?

Line 480 – how is the Bhutanese government committed – what evidence can support this as ‘appears committed’ seems a little vague

Your strengths / limitations and conclusion sections all make good sense for me.

All the best as you finalise this work.

6. PLOS authors have the option to publish the peer review history of their article (what does this mean?). If published, this will include your full peer review and any attached files.

**Do you want your identity to be public for this peer review?** Reviewer #1: **Yes: **Dr. Maha Atout. Philadelphia University. Amman. Jordan

Reviewer #2: **Yes: **Dr Claudia Virdun

---

## [Decision Letter · Decision Letter 1]

27 Oct 2022

Healthcare professionals’ views on how palliative care should be delivered in Bhutan: a qualitative study

PGPH-D-22-00738R1

Dear Dr. Laabar

We are pleased to inform you that your manuscript 'Healthcare professionals’ views on how palliative care should be delivered in Bhutan: a qualitative study' has been provisionally accepted for publication in PLOS Global Public Health.

Best regards,

Carmen García Peña, PhD

Academic Editor

Excellent work. I suggest just to double check that all the supplementary files do not contain data that could identify the participants

Reviewer Comments (if any, and for reference):

Reviewer's Responses to Questions

**Comments to the Author**

1. If the authors have adequately addressed your comments raised in a previous round of review and you feel that this manuscript is now acceptable for publication, you may indicate that here to bypass the “Comments to the Author” section, enter your conflict of interest statement in the “Confidential to Editor” section, and submit your "Accept" recommendation.

Reviewer #2: All comments have been addressed

2. Does this manuscript meet PLOS Global Public Health’s publication criteria? Is the manuscript technically sound, and do the data support the conclusions? The manuscript must describe methodologically and ethically rigorous research with conclusions that are appropriately drawn based on the data presented.

Reviewer #2: Yes

3. Has the statistical analysis been performed appropriately and rigorously?

Reviewer #2: N/A

4. Have the authors made all data underlying the findings in their manuscript fully available (please refer to the Data Availability Statement at the start of the manuscript PDF file)?

Reviewer #2: Yes

5. Is the manuscript presented in an intelligible fashion and written in standard English?

Reviewer #2: Yes

6. Review Comments to the Author

Reviewer #2: Thanks - you have addressed all comments well and I think this is an excellent contribution to the literature. All the best as you continue this important work.

7. PLOS authors have the option to publish the peer review history of their article (what does this mean?). If published, this will include your full peer review and any attached files.

**Do you want your identity to be public for this peer review?** For information about this choice, including consent withdrawal, please see our Privacy Policy.

Reviewer #2: **Yes: **Claudia Virdun, Senior Research Fellow, Queensland University of Technology
